# Periodontal health status in systemic sclerosis patients: Systematic review and meta-analysis

**Stefan Sredojevic**[1]*, **Dejana Colak**[2], **Rok Gaspersic**[2], **Slavica Pavlov Dolijanovic**[3], **Aleksandar Jakovljevic**[4], **Natasa Nikolic-Jakoba**[1]

**1** Department of Periodontology, School of Dental Medicine, University of Belgrade, Belgrade, Serbia, **2** Department of Oral Diseases and Periodontology, Dental Clinic, University Medical Centre Ljubljana, University of Ljubljana, Ljubljana, Slovenia, **3** Institute of Rheumatology, School of Medicine, University of Belgrade, Belgrade, Serbia, **4** Department of Pathophysiology, School of Dental Medicine, University of Belgrade, Belgrade, Serbia

* sredojevic43@gmail.com

## Abstract

### Objective

The present systematic review and meta-analysis aimed to evaluate the periodontal health of systemic sclerosis patients compared with non-systemic sclerosis controls.

### Materials and methods

Online databases were searched for eligible studies on February 24, 2023. The primary outcomes of interest in systemic sclerosis patients and controls included the clinical attachment level, periodontal probing depth, recession depth, plaque index, bleeding on probing score, gingival index, number of teeth with periodontitis, prevalence of periodontitis and gingivitis, and extent and severity of periodontitis.

### Results

Fourteen studies met inclusion criteria and were incorporated in the qualitative and quantitative analysis. In comparison with the controls, systemic sclerosis patients had a higher prevalence of periodontitis (OR = 7.63 (1.74–33.50), p = 0.04, $I^2$ = 69%), including more severe forms of periodontitis (OR = 6.68 (3.39–13.15), p = 0.85, $I^2$ = 0%), as well as higher periodontal probing depth ((0.88 (0.45–1.31), p = 0.02, $I^2$ = 99%)), clinical attachment level (1.22 (0.8–1.64), p = 0.003, $I^2$ = 98%), and plaque presence (0.83 (0.13–1.53), p = 0.03, $I^2$ = 96%). There was no statistically significant difference in gingival inflammation (1.14 (0.07–2.21), p = 0.04, $I^2$ = 98%).

### Conclusions

The systematic review and the meta-analysis showed that systemic sclerosis patients suffer from worse periodontal health than non-systemic sclerosis individuals.

**Data Availability Statement:** All relevant data are within the paper and Supporting Information files.

**Funding:** This research has been supported by Ministry of Education, Science and Technological Development of Republic of Serbia [Grants No. 451-03-68]. The funders had no role in study design, data collection and analysis, decision to publish, or preparation of the manuscript. The authors received no specific funding for this work.

**Competing interests:** The authors have declared that no com peting interests exist.

## Introduction

Systemic sclerosis (SSc) is a chronic rheumatic inflammatory disease of unknown origin that is characterized by extensive fibrosis of the skin and internal organs, vasculopathy, and immune dysfunction [1]. SSc has been reported to be three times more prevalent in females than in males, with frequency peaks in the sixth and seventh decades of life [2]. This systemic disease is associated with higher mortality rates compared to other rheumatic diseases with a five-year cumulative survival rate of 75% [3]. The hallmark of SSc is microvascular damage which triggers immune cells to produce autoantibodies as well as proinflammatory and profibrotic cytokines. This cascade of events leads to excessive collagen production by means of activated fibroblasts and myofibroblasts, thus resulting in collagen accumulation in the skin and internal organs [4]. Although most inflammatory and fibrotic changes are detected on the skin, it is fibrosis of internal organs, primarily the lungs, heart, and kidneys, that causes severe morbidity in these patients [5]. Based on the degree of skin involvement, SSc can be divided into two subgroups–limited cutaneous (lSSc) and diffuse cutaneous systemic sclerosis (dSSc). The latter is more severe and progresses more rapidly [6]. Despite treatment options being available for many aspects of systemic sclerosis, there are numerous non-fatal consequences that reduce SSc patients' quality of life [7]. Almost 80% of people with SSc will experience some orofacial symptoms.This spectrum encompasses microstomy, hyposalivation, dysphagia, caries lesions, and presumably periodontal diseases [8].

Within the new etiopathogenesis-based classification of periodontal diseases and conditions, systemic sclerosis (scleroderma) has been assigned to the category of systemic diseases that can have an impact on periodontal tissue destruction [9]. As a chronic inflammatory disease, periodontitis shares several pathogenetic characteristics with scleroderma. Common pathogenetic features include prominent acute inflammation during initial stages, followed by inflammatory-mediated tissue atrophy and destruction in the advanced stages [10]. Even though a few prior studies [11–13] have reported that periodontal destruction observed in SSc patients was more severe when compared to systemically healthy subjects, other investigations have not been able to confirm this finding [14, 15]. Apart from the known factors that may contribute to periodontal destruction in SSc patients, such as increased plaque acummulation, it is likely that there are additional underlying mechanisms involved and these require further examinations [16]. Two recent studies demonstrated a unique periodontal profile in individuals with SSc, characterized by increased prevalence of gingival recession and reduced bleeding on probing [11, 12]. Therefore, it is difficult to diagnose periodontal disease in its earliest stages in patients with SSc due to diminished bleeding upon probing [11]. The functional decline of the masticatory system and concomitant dysphagia may occur in SSc patients due to increased tooth mobility and tooth loss in advanced stages of periodontitis [17]. Therefore, it is of utmost importance to assess the risk of periodontitis development in SSc patients in order to establish periodontal screening protocols.

Therefore, the objective of the present systematic review was to evaluate whether SSc patients are more susceptible to periodontitis than non-SSc individuals. To the best of our knowledge, this systematic review, along with the meta-analysis, represents the first qualitative and quantitative investigation focused entirely on the periodontal status of SSc patients.

## Methods

The Preferred Reporting Items for Systematic Reviews and Meta-Analyses (PRISMA) Statement from 2020 [18] was followed for the purpose of this review (S1 Table, PRISMA 2020 checklist). The review protocol was registered at the International Prospective Register of Systematic Reviews (PROSPERO) under the record number CRD42021266901. The Population/

Intervention/Comparison/Outcome (PICO) question was set up as follows: Do systemic sclerosis patients have worse periodontal health in comparison with non-systemic sclerosis individuals?

## Search strategy

On February 24, 2023, the electronic search approach was applied to Clarivate Analytics' Web of Science, Scopus, PubMed (including MEDLINE), and CENTRAL (Cochrane Central Register of Controlled Trials) with no language restriction (S2 Table). The keywords and MeSH headings for "systemic sclerosis" and "periodontitis" were used in the search approach. A hand search of international journals whose scope is within the interest of the current study was carried out with the same keywords and phrases. Thereby, *Rheumatology*, *Clinical Rheumatology*, *Journal of Scleroderma and Related Disorders*, *Journal of Clinical Periodontology*, *Journal of Periodontology*, *Journal of Periodontal Research*, along with the reference lists of pertinent studies, were included. Full-text studies, including early-view articles, were taken into consideration. In addition, a hand search of grey literature was performed using GreyLit (www.greylit.org). The results generated upon the literature search were downloaded and imported to Rayyan [19]. Next, Rayyan's duplicate identification strategy was applied to automatically remove any duplicates, which was afterwards also conducted manually by the authors (A.J. and R.G.).

## Population of interest

The population of interest in the systematic review were adult individuals (>18 years) with a diagnosis of SSc. The primary prerequisite was that the patients had been diagnosed with SSc in accordance with the European League Against Rheumatism (EULAR)/American College of Rheumatology (ACR) classification criteria based on skin sclerosis, vascular changes (i.e., a history of Raynaud's phenomenon and nailfold capillary destruction) and/or the presence of specific antibodies (i.e., anticentromere antibodies and anti-topoisomerase I antibodies) [20]. The controls in the studies were adult individuals (>18 years) without SSc whose age and gender matched with the SSc patients.

## Inclusion study criteria

To answer the main research question of the current systematic review, we included randomized clinical trials, case-control and cross-sectional studies conducted on SSc patients that contained data on periodontal clinical parameters. Study exclusion criteria were as follows: studies without a control group, case reports, reviews (narrative, scope and/or systematic) with or without a meta-analysis, unpublished studies (e.g., conference abstracts, trial protocols, etc.), animal studies, and studies without proper clinical periodontal examinations or data on SSc.

## Study selection process

Two independent reviewers (S.S., D.C.) selected the studies by screening the titles and abstracts generated by database and hand search strategies. The Rayyan software was used to identify duplicate records and inappropriate publication types [19]. The full text of the articles was screened when needed. At this point, the excluded studies were grouped according to the reason for their exclusion, as presented in S3 Table.

## Data collection process

The two authors (S.S., D.C.) independently pooled relevant data from the included studies. The data was tabularized into a predetermined table. The information collected from the

included studies comprised the lead author, year of publication, country in which the study was conducted, demographic characteristics of the case and control groups, sample size, periodontal outcomes recorded, SSc outcomes measured, main study findings, limitations, and details regarding the reported funding source.

## Main study outcomes

The outcomes of interest in SSc and control cohorts pertained to the following clinical periodontal parameters: the clinical attachment level (CAL), periodontal probing depth (PPD), gingival recession (REC), plaque index (PI), full mouth bleeding on probing score (BOP), gingival index (GI), Community Periodontal Index Treatment Needed (CPITN), furcation involvement, number of teeth with periodontitis, periodontitis and gingivitis prevalence (%), periodontitis extent and severity.

## Risk of bias (quality) assessment

Two reviewers (S.S., D.C.) independently evaluated the quality of the cross-sectional studies. Any disagreements were resolved by the third reviewer (R.G.). The quality of included publications was assessed using original questions from the National Heart, Lung, and Blood Institute's (NHLBI) Quality Assessment Tool for Observational Cohort and Cross-Sectional Studies [21]. The estimations of the following factors were performed: the target population, case and control representation, sample size rationale, groups drawn from the same population, main research question, inclusion and exclusion criteria, definitions of case and control cohorts, randomness of the cases and controls, concurrent controls, exposure assessed prior to outcome measurement, and statistical analysis. The responses (yes, no, cannot decide, not reported, not applicable) were offered for each question. Based on these ratings, each study was classified as good, fair, or poor (S4 Table). The meta-analysis did not include low-quality studies (rated as poor).

## Effects measured

The main outcome of the systematic review and the meta-analysis was the mean difference between periodontal clinical parameters (i.e., the number of teeth with periodontitis, PPD, CAL, gingival indices, plaque indices, and other periodontal indices) and the odds ratio (OR) of periodontitis prevalence between SSc patients and non-SSc controls.

## Strategy for data synthesis

The findings from the included studies were summarized in the systematic review and, when applicable, in the meta-analysis. A meta-analysis was performed when at least two studies reported data on the same periodontal outcome. The synthesis was performed using the statistical software Review Manager version 5.3. The OR was estimated for dichotomous outcomes (e.g., the prevalence of periodontitis and severe periodontitis). As for continuous outcomes (CAL, PPD, % PPD > 4mm, gingival and plaque indices), the mean differences were calculated if studies employed the same parameters to quantify the outcome of interest. Nevertheless, if other indices were used, standardized mean differences were calculated. The inverse Variance method was used when the measurements tested had a continuous outcome, whereas the Mantel-Haenszel method was used for dichotomous outcomes. When the data for the same outcome were presented differently (e.g., the mean in one study, the median in another), the statistical approaches recommended by the Cochrane Reviewers' Handbook were used to convert the medians into means to allow data combining [22–24]. The findings were summarized in a meta-analysis using the random effects model. The heterogeneity among studies was

assessed using the $I^2$ value, where $I^2 < 25\%$ was taken as an indicator of low heterogeneity, while $I^2 > 75\%$ implied high heterogeneity [24]. The presence of asymmetry in the funnel plots was interpreted as the potential presence of reporting bias. The p-value and 95% confidence interval for each relevant outcome were displayed. For all analyses, the alpha was set at 0.05. The subgroup analysis was performed for two forms of SSc (limited and diffuse) when sufficient data was available.

## Results

### Selection and characteristics of the included studies

The search strategies identified 708 potentially relevant publications. After discarding duplicates, 426 studies remained and were screened for eligibility. Following the initial screening of the titles and abstracts, 397 studies were excluded. Therefore, the full text of 25 studies was evaluated in detail, with 11 studies being excluded from the review on the basis stated in the PRISMA flow diagram (Fig 1).

Finally, a total of 14 studies fitted the inclusion criteria and they were combined into the qualitative and quantitative systematic review [11–15, 25–33]. The main characteristics of the included studies are listed in Table 1. Owing to the a priori-defined eligibility criteria, there were no disagreements between the independent reviewers in the process of article selection and inclusion. Additionally, a very high percentage of agreement (i.e., 95%) between the independent reviewers

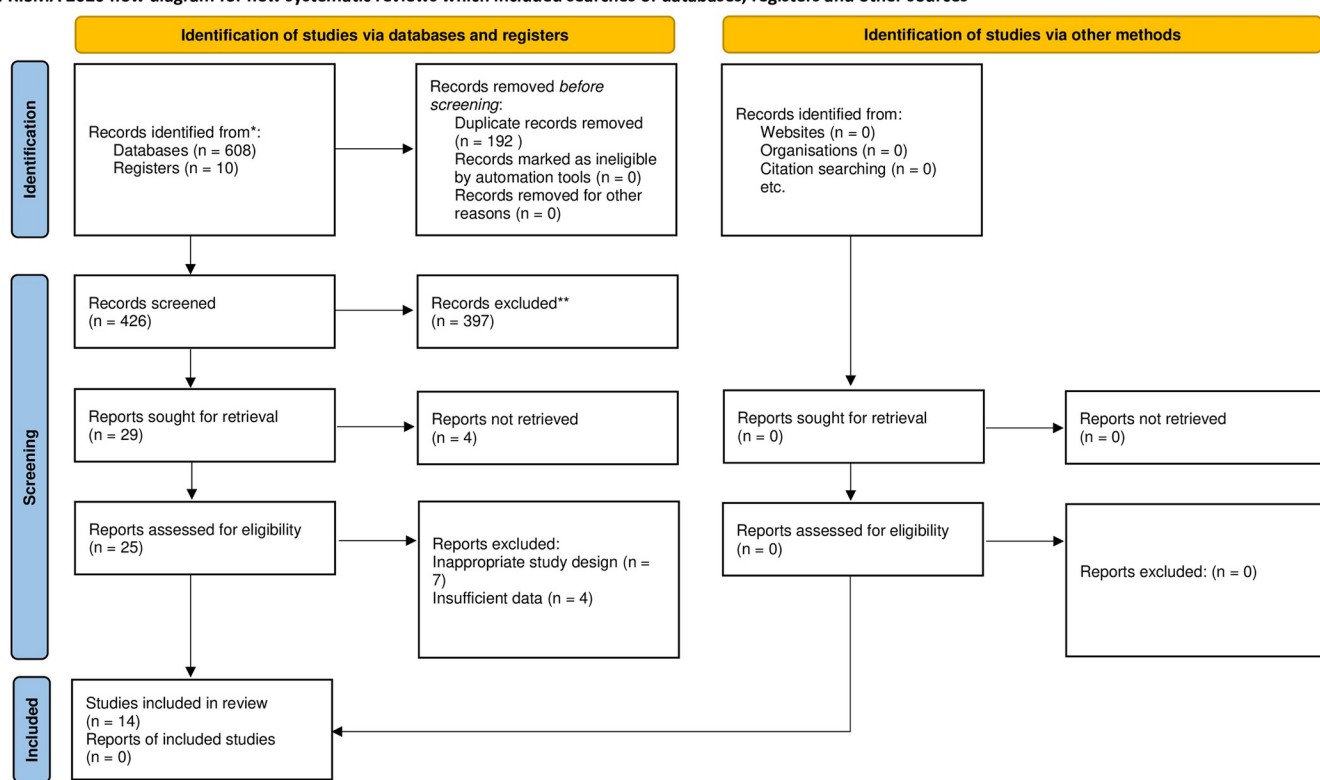

**PRISMA 2020 flow diagram for new systematic reviews which included searches of databases, registers and other sources**

*Consider, if feasible to do so, reporting the number of records identified from each database or register searched (rather than the total number across all databases/registers).
**If automation tools were used, indicate how many records were excluded by a human and how many were excluded by automation tools.

*From:* Page MJ, McKenzie JE, Bossuyt PM, Boutron I, Hoffmann TC, Mulrow CD, et al. The PRISMA 2020 statement: an updated guideline for reporting systematic reviews. BMJ 2021;372:n71. doi: 10.1136/bmj.n71. For more information, visit: http://www.prisma-statement.org/

**Fig 1. PRISMA 2020 flow diagram.**

**Table 1. Characteristics of the studies included in the systematic review.**

| Author, year | Country | Study sample | Average age (SSc/ controls) | Diffuse type of SSc (%) | Female (SSc/ controls) | Average SSc duration | Periodontal parameters assessed | Main findings | Limitations | Reported funding |
|---|---|---|---|---|---|---|---|---|---|---|
| Wood et al., 1988 | Canada | SSc (n = 31) Nonrheumatic disease controls (n = 30) | 51.9/ 50.3 | 51.6 | 31/30 | 7.5 | Plaque score, calculus score, gingivitis score, pocketing score | SSc patients had higher PPD and GI than control group. | Small sample size | Reported |
| Chu, 2011 | China | SSc (n = 42) Nonsystemic sclerosis controls (n = 42) | 54/ NA | NA | 41/NA | NA | CPITN | SSc patients had similar periodontal status compared to controls. | Selection of controls | Reported |
| Leung, 2011 | China | SSc (n = 36) Systemically healthy controls (n = 36) | 50.6/ NA | NA | 36/NA | 12.3 | PI, BOP, PPD, CAL | SSc patients had higher PPD and BOP compared to controls. | Small sample size | Reported |
| Matarese, 2012 | Italy | SSc (n = 30) Systemically healthy periodontitis controls (n = 30) Systemically healthy nonperiodontitis controls (n = 30) | 52.4/ NA/ NA | NA | NA | 19 | NA | PI, BOP, PPD, CAL, CPITN | SSc patients had higher CAL and PPD compared to controls. | Small sample size | Not reported |
| Mayer, 2013 | Israel | SSc (n = 12) Systemically healthy periodontitis controls (n = 12) | 48.4/ 51.5 | NA | 9/7 | 7.2 | PPD, FMBS, PI, GI, No. of sites with PPD>4 mm | SSc patients had a higher prevalence of periodontitis and higher TNF-α levels in gingival fluid than controls. | Small sample size | Not reported |
| Baron, 2014 | Canada | SSc (n = 163) Systemically healthy controls (n = 231) | 56.2/ 58.1 | 28.3 | 146/209 | 13.7 | CAL, PPD, number of teeth with periodontal disease | SSc patients had a larger number of teeth with periodontal disease than controls. | The study sample was not severely involved | Reported |
| Elimelech, 2015 | Israel | SSc (n = 20) Systemically healthy controls (n = 20) | 45.36/ 48.5 | 50 | 18/15 | 7.2 | PPD, BOP, GI, PI | SSc patients had higher BOP and GI and higher TNFα level in GCF as compared to the controls. | Small sample size | Not reported |
| Pischon, 2016 | Germany | SSc (n = 58) Systemically healthy controls (n = 52) | 55.1/ 52.1 | 62 | 44/43 | 6.2 | PI, GI, BOP, PPD, CAL | SSc patients had higher CAL, but lower GI than controls. | High number of the diffuse type of SSc, selection of controls | Not reported |
| Isola, 2017 | Italy | SSc (n = 54) Systemically healthy controls (n = 55) | 48.7/ 47.3 | 79.6 | 36/30 | 7.4 | PI, GI, BOP, CAL, PPD, Percentage of sites with PPD≥4 mm | SSc patients had higher CAL compared to controls. | Cross sectional study design | Reported |
| Iordache, 2019 | Romania | SSc (n = 43) Systemically healthy controls (n = 43) | 43.95/ NA | 67.7 | 31/NA | 8.7 | PI, GI, BOP, PPD, Teeth with periodontal disease/ percentage of sites with CAL≥5.5 mm | SSc patients had higher PPD and more severe forms of periodontitis compared to controls. | Small sample size | Not reported |

*(Continued)*

**Table 1.** (Continued)

| Author, year | Country | Study sample | Average age (SSc/controls) | Diffuse type of SSc (%) | Female (SSc/controls) | Average SSc duration | Periodontal parameters assessed | Main findings | Limitations | Reported funding |
|---|---|---|---|---|---|---|---|---|---|---|
| Da Silva, 2019 | Brazil | SSc (n = 50) Systemically healthy controls (n = 43) | 46/ 44 | 52 | 43/36 | NA | GBI, PPD, CAL, GR, severity and extension of periodontitis | SSc patients presented with a high prevalence of periodontal disease, higher GR and lower BOP than controls. | Small sample size | Reported |
| Polizzi 2020 | Italy | SSc (n = 70) Systemically healthy controls (n = 75) | 53.2/ 51.4 | 72.3 | 42/39 | 7.6 | PI, BOP, PPD, CAL, percentage of sites with CAL≥4 mm | SSc patients had higher CAL compared to controls. | Cross sectional study design | Reported |
| Isola 2021 | Italy | SSc nonperiodontitis (n = 36) SSc periodontitis (n = 35) Systemically healthy periodontitis (n = 37) Systemically healthy nonperiodontitis (n = 37) | 53.2/ 52.5/ 52.7/ 52.9 | 76.2/ 73.2 | 20/20/ 18/19 | NA | CAL, BOP, PI, sites with CAL 4–5 mm/sites with CAL≥6 mm/sites with PPD 4–5 mm/ sites with PPD≥6 mm | SSc patients had a lower number of teeth and higher PPD, CAL, and BOP levels. | Cross sectional study design | Not reported |
| Buchbender 2021 | Germany | SSc (n = 17) Systemically healthy controls (n = 22) | 62.2/ 61.5 | 35.2 | 12/15 | NA | PI, BOP, CAL, PPD, GR | SSc patients had higher CAL, BOP and GR compared to controls. | Small sample size | Reported |

Abbreviations: SSc: systemic sclerosis; CAL: clinical attachment level; PPD: periodontal probing depth; GR: gingival recession; GI: gingival index; PI: plaque index; BOP: bleeding on probing; CPITN: community periodontal index of treatment needs; GBI: gingival bleeding index; NA: not applicable

was observed during the process of quality assessment of the articles included in the final review. Furthermore, any disagreements were resolved by another author (R.G.).

The earliest study included in the review was published in 1988 [27], while the most recent article was published in 2021 [33]. Seven studies were conducted in Europe (Italy, Romania, Germany); two studies each were conducted in China, Canada and Israel and there was one study conducted in Brazil. All included publications employed the case-control design. The included studies comprised 626 patients with SSc and 691 controls. The controls in all studies were pooled from the general population. All the studies presented data on clinical periodontal parameters and most of them contained some SSc clinical parameters.

## Quality assessment of the included studies

Quality assessment of the included studies is presented in S4 Table. The majority of the studies were rated as good [11, 13, 15, 16, 28–33], while others were rated as fair [12, 14, 25, 27]. Sample size justification and statistical adjustment of critical potentially confounding variables posed the most significant concern for bias across all included studies. Only one study justified the sample size to ensure that the study was adequately powered to detect an association [29].

## Descriptive results from the included studies

Of these 14 articles, three [14, 15, 32] reported on the prevalence, extent, and severity of periodontitis in patients with SSc, whereas others used continuous variables, including PPD, CAL,

and BOP. All the included studies utilized manual periodontal probes for conducting periodontal examinations. Specifically, the PCP 15 periodontal probe (Hu-Friedy, Chicago, Illinois, USA) was predominantly used, i.e., in six studies [11, 13, 29, 30, 32]; three studies employed the Community Periodontal Index (CPI) probe [14, 15, 27], and one study each used the Williams PCP 2 [16], the PCP 11 [12] and the PCP 12 [33] probes. The specific type of manual probe used in the remaining studies was not reported. In all studies, the clinical examiners were described as experienced. Examiner calibration was performed in the majority of the included studies [13, 16, 25, 28–31, 33] to ensure consistency and accuracy of the assessment.

Iordache et al. found that more than half of the patients in the SSc group had periodontal disease [32]. The presence of periodontitis in this study was defined as either PPD > 3 mm or CAL ≥ 2 mm. Two recent studies have also observed that periodontitis is more common in SSc patients, especially its severe and generalized forms [11, 12]. For the severity and extent of periodontitis, Pischon used the classification by Eke et al. [34], while Da Silva applied the American Academy of Periodontology 2015 classification [35]. Most of the remaining studies showed that patients with SSc had higher values of periodontal parameters (CAL and PPD) than systemically healthy individuals. Only one study reported that the periodontal health status of patients with SSc and systemically healthy individuals was similar [14]. Of the 14 included studies, seven found higher levels of gingival inflammation in patients with SSc [13, 15, 27–29, 31, 33], while two studies showed that gingival inflammation scores were significantly reduced in patients with SSc [11, 12]. There was a diversity of periodontal indices used for the assessment of gingival inflammation in the studies. Of the nine studies that assessed gingival inflammation, five used the Löe-Silness Gingival Index [11, 13, 25, 28, 32], three only used the BOP score [15, 31, 33], and one used the Carter and Barnes Gingival Index [12]. Eight of the eleven studies evaluating the presence of plaque used the PI according to Silness and Loe [11, 13, 25, 28–32], two studies used the dichotomous plaque score [15, 27], and one used the Mombelli Plaque Index [33]. The results of two recent studies showed that the degree of a number of periodontal parameters was strongly associated with the severity and duration of SSc [13, 29]. Two studies did not report on the duration of SSc [14, 31] and four studies did not include the subgroup analysis of different forms of SSc [14, 15, 28, 31]. Additionally, most studies did not report on the activity and severity of SSc.

## Meta-analysis of clinical periodontal parameters in systemic sclerosis patients and controls

The results of the meta-analysis of clinical periodontal parameters in SSc patients and controls are shown in Fig 2.

The meta-analysis showed SSc patients had a higher prevalence of periodontitis and more severe periodontal destruction. Also, they exhibited higher PPD values (0.88 (0.45–1.31), $p = 0.02$, $I^2 = 99\%$), more sites with PPD>4mm (0.97 (0.75–1.18), $p = 0.36$, $I^2 = 9\%$)), CAL (1.22 (0.8–1.64), $p = 0.003$, $I^2 = 98\%$), and increased plaque levels (0.83 (0.13–1.53), $p = 0.03$, $I^2 = 96\%$). Additionally, there was no statistically significant difference in gingival inflammation (1.14 (0.07–2.21), $p = 0.04$, $I^2 = 98\%$) and the number of missing teeth (2.59 (0.01–5.19), $p = 0.05$, $I^2 = 97\%$). The heterogeneity among included studies in terms of CAL, PPD, gingival inflammation, plaque presence, the number of missing teeth, and the prevalence of periodontitis was high ($I^2$ 69–98%). The studies included in the analysis of the prevalence of moderate-severe periodontitis and PPD>4mm did not present with heterogeneity. The funnel plot for the meta-analysis of PPD showed asymmetry and reporting bias. Regarding the meta-analysis of PPD, the studies with a smaller number of participants showed little or no difference among

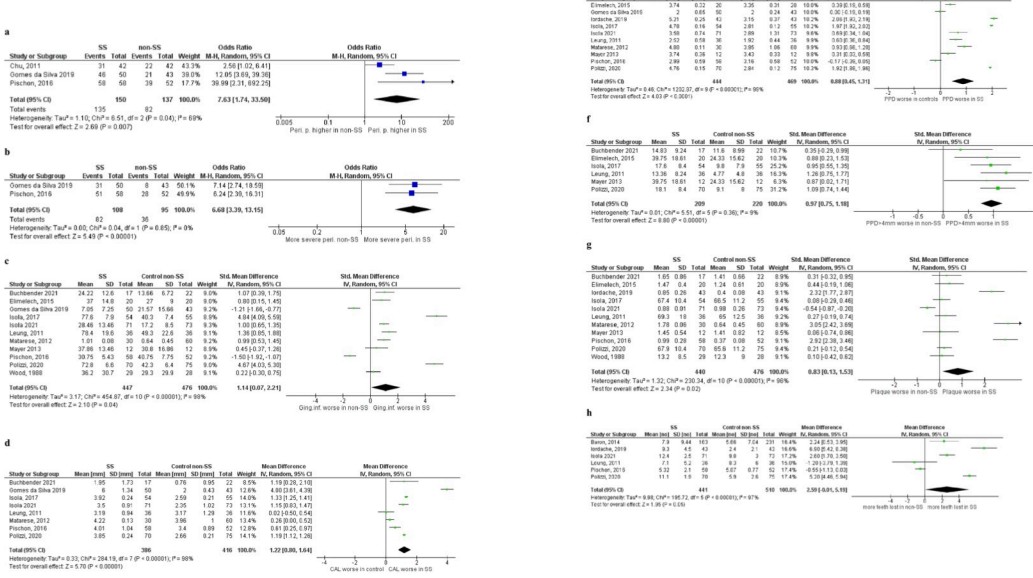

**Fig 2. Meta-analysis of periodontal parameters in systemic sclerosis patients and controls. a** prevalence of periodontitis; **b** prevalence of moderate to severe periodontitis vs mild and no periodontitis; **c** gingival inflammation; **d** CAL; **e** PPD; **f** sites with PPD>4mm; **g** plaque indices; **h** number of missing teeth.

SSc patients and controls, while the more extensive studies showed otherwise. The estimated effect of SSc on PPD might be more significant than indicated by the meta-analysis. The funnel plot for the meta-analysis focusing on the differences in gingival inflammation levels between SSc patients and controls showed satisfactory symmetry with only a few studies showing scattering from the estimated effect size. The funnel plot for the meta-analysis of dental plaque presence showed asymmetry with most of the larger studies demonstrating no difference in plaque levels between SSc patients and controls. However, a few small-sized studies found significant differences between the two groups. As a consequence, the certainty of evidence obtained by the meta-analysis of plaque presence was diminished. The funnel plots of the study effect size in the meta-analysis of the periodontal parameters are presented in S1 Fig.

## Discussion

The current systematic review sought to compare the periodontal health status of SSc patients to that of non-SSc individuals. According to the findings, patients with SSc have poorer periodontal health than non-SSc controls. The meta-analysis revealed that periodontitis was more prevalent among SSc patients, accompanied by moderate to severe periodontal destruction. Compared to non-SSc controls, SSc patients had higher clinical periodontal parameters such as PPD, CAL, and dental plaque indices. However, there was no statistically significant difference in gingival inflammation or tooth loss between SSc and non-SSc individuals. In addition, most of the included studies reported that CAL was higher in patients with SSc compared with systemically healthy controls [11–13, 29, 31]. Among the studies that found increased CAL in patients with SSc, two studies statistically adjusted for potential risk factors, including age, gender, smoking status, education, and alcohol consumption [11, 26]. In contrast, the study by Chu et al. did not observe higher periodontal destruction in patients with SSc compared with individuals without SSc [14]. This finding could be explained by the fact that the control patients were selected from the patients who already required periodontal treatment at the

dental clinic, implying that they had worse periodontal health than the general population. Despite this, most patients in both SSc and control groups required advanced periodontal treatment.

Plaque accumulation may increase in SSc patients due to limited mouth opening, decreased manual dexterity, and hyposalivation [36]. Pischon et al., on the other hand, discovered increased periodontal destruction in SSc patients compared to control subjects, which remained statistically significant after adjusting for plaque accumulation [11]. This finding supported previous research that emphasized the multifactorial etiology of periodontitis in SSc patients [16].

The findings revealed that there was no consistency in terms of gingival inflammation levels between patients with SSc and systemically healthy controls. Although gingival inflammation is thought to be a prerequisite for subsequent clinical attachment loss in periodontitis [37], two recent studies showed reduced gingival inflammation scores along with higher CAL in SSc patients [11, 12]. This discrepancy between increased periodontal destruction and suppressed clinical signs of gingival inflammation may be explained by microvascular abnormalities and chronic fibrosis observed in the gingiva of SSc patients [38, 39]. On the other hand, corticosteroids and immunosuppressive drugs used in the treatment of SSc are known to be able to mask gingival inflammatory signs, such as edema and bleeding on probing [4, 40]. Furthermore, none of the studies included patients in the early stages of SSc, i.e., prior to receiving treatment.

Rheumatic diseases and periodontitis are complex chronic diseases with a common etiopathogenetic pathway characterized by inflammatory-mediated tissue destruction and fibrosis, eventually leading to loss of function [41]. A recent study has reported that cytokines such as TGF-β1 and VEGF may be involved in developing an immune response in periodontitis and SSc [31]. Furthermore, higher levels of proinflammatory mediators such as TNFα were found in the gingival fluid of SSc patients [25]. In addition, the receptor activator of NF-κB ligand (RANKL), one of the major regulators of bone resorption in periodontitis, was significantly higher in the serum of SSc patients than in the controls [42, 43]. Although the link between the two diseases has not yet been fully established, research suggest that poor oral hygiene, proinflammatory cytokines, and microvascular alterations in SSc patients may promote the onset and progression of periodontitis [26, 36, 44].

The current systematic review has certain limitations because of: (i) the small number of eligible studies; (ii) the small and unjustified sample size of several included studies; (iii) the high heterogeneity of included studies; and (iv) all included studies employing the case-control design. The small sample size of several eligible studies may raise the margin of error and result in low power. Furthermore, due to the small sample size, we were unable to perform the subgroup analysis on SSc patients with diffuse and limited forms. It is noteworthy that most studies included in the present systematic review did not mention the power of the study, but due to the rarity of SSc, this is not considered a 'fatal flaw'. For most periodontal parameters, the high heterogeneity among the studies included in the meta-analysis suggests discrepancies between individual study results. This could stem from the fact that patients with different stages of SSc were included, which could have resulted in varying degrees of periodontal destruction being observed. In the studies, the selection of controls was also based on various criteria. Although systemically healthy individuals were selected for the control group in most studies, individuals without SSc were selected in the study by Wood [27]; and individuals without rheumatic disease were selected in the study by Chu [14]. In addition, the control groups in the two studies consisted of systemically healthy patients with periodontitis [19, 23]. The use of different indices to determine the same parameters when it comes to gingival inflammation and dental plaque, as well as different diagnostic criteria for assessing periodontitis,

contributed to the high heterogeneity among the included studies. What is more, all included studies used case-control designs, therefore, the cause-effect relationship between SSc and periodontitis could not be fully elucidated.

Besides the use of validated research techniques for searching electronic databases, the processes of selection and quality assessment of included studies are among the review's strengths. Another advantage of the current systematic review is that it combined the findings of studies of high methodological quality. Eleven of the fourteen included studies were rated as high quality, while the others were rated as fair. In addition, the meta-analysis considered the high heterogeneity when opting for the appropriate statistical test. Despite the considerable heterogeneity, the presence of a statistical difference between the SSc and control groups implies that SSc patients have poorer periodontal health. This finding is consistent with previous articles on the subject [8, 45].

The present review highlights the importance of regular screening for periodontitis in SSc patients to detect early signs of periodontal disease and prevent its progression. SSc patients often receive care from multiple specialists, including rheumatologists and periodontists. This systematic review emphasizes the need for collaboration between these specialists to ensure that patients receive comprehensive care that addresses both their systemic sclerosis and periodontal health. Moreover, educating patients on the connection between periodontal health and systemic health may motivate them to prioritize their oral health and improve their overall health outcomes. The relationship between SSc and periodontitis should be investigated further since periodontitis represents a known risk factor for low-grade systemic inflammation and some systemic diseases [46, 47].

## Conclusion

The current systematic review and meta-analysis findings suggest that periodontitis is more prevalent in patients with SSc. Furthermore, higher PPD, CAL, and PI values have been found in SSc patients than in non-SSc individuals. Early detection of periodontal disease in SSc patients is critical for receiving early treatment and avoiding invasive periodontal procedures as the disease progresses, since this is when severe microstomia may occur. This highlights the periodontist's critical role in diagnosing and treating SSc as part of a well-coordinated multidisciplinary team. Further research with a higher level of evidence is needed to clarify the relationship between the two diseases and establish protocols and recommendations for periodontal treatment of SSc patients.

## Supporting information

**S1 Fig. Funnel plots of study effect size in meta-analysis of the periodontal parameters. a** gingival inflammation; **b** PPD; **c** PPD >4 mm; **d** plaque indices; **e** CAL.
(TIF)

**S1 Table. PRISMA 2020 checklist.**
(DOCX)

**S2 Table. Electronic databases and search strategy [February 24, 2023].**
(DOCX)

**S3 Table. The study excluded during full-text screening.** A—Inappropriate study design; B —Insufficient data.
(DOCX)

**S4 Table. Quality assessment of included studies.** *CD, cannot determine; NA, not applicable; NR, not reported.
(DOCX)

# Acknowledgments

We would like to express our gratitude to Jelena Jacimovic, a librarian at the School of Dental Medicine, University of Belgrade, for performing the electronic search and to Irena Aleksic-Hajdukovic, Assistant Professor in English Language at the School of Dental Medicine, University of Belgrade, for improving the language of the manuscript.

# Author Contributions

**Conceptualization:** Stefan Sredojevic, Dejana Colak, Rok Gaspersic, Slavica Pavlov Dolijanovic, Aleksandar Jakovljevic, Natasa Nikolic-Jakoba.

**Investigation:** Stefan Sredojevic, Dejana Colak, Aleksandar Jakovljevic.

**Methodology:** Stefan Sredojevic, Dejana Colak, Slavica Pavlov Dolijanovic.

**Supervision:** Rok Gaspersic.

**Validation:** Natasa Nikolic-Jakoba.

**Visualization:** Slavica Pavlov Dolijanovic, Aleksandar Jakovljevic, Natasa Nikolic-Jakoba.

**Writing – original draft:** Natasa Nikolic-Jakoba.

**Writing – review & editing:** Rok Gaspersic, Aleksandar Jakovljevic, Natasa Nikolic-Jakoba.

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
