## [Decision Letter · Decision Letter 0]

20 Jun 2023

PONE-D-23-12300

Periodontal health status in systemic sclerosis patients: systematic review and meta-analysis

PLOS ONE

Dear Dr. Sredojevic,

Thank you for submitting your manuscript to PLOS ONE. After careful consideration, we feel that it has merit but does not fully meet PLOS ONE’s publication criteria as it currently stands. Therefore, we invite you to submit a revised version of the manuscript that addresses the points raised during the review process.

We look forward to receiving your revised manuscript.

Kind regards,

Stefano Corbella, DDS, PhD

Academic Editor

PLOS ONE

Journal Requirements:

"The funders has no role in study design, data collection and analyses, decision to publish, or preparation of the manuscript."

Additional Editor Comments :

Please consider ALL the reviewers' comments before resubmitting the paper.

Reviewers' comments:

Reviewer's Responses to Questions

**Comments to the Author**

1. Is the manuscript technically sound, and do the data support the conclusions?

Reviewer #1: Partly

Reviewer #2: Yes

2. Has the statistical analysis been performed appropriately and rigorously? 

Reviewer #1: No

Reviewer #2: Yes

3. Have the authors made all data underlying the findings in their manuscript fully available?

Reviewer #1: Yes

Reviewer #2: Yes

4. Is the manuscript presented in an intelligible fashion and written in standard English?

Reviewer #1: Yes

Reviewer #2: Yes

5. Review Comments to the Author

Reviewer #1: 1. Please edit language throughout the text where needed2. Reg your PICO: impaired periodontal condition isnot properly articulated, perhaps you need to rephrase to worse instead of impaired. 3. Abbreviations of your main outcomes do not align with the outcomes, FMPS instead of PI and also references are missing.4. The agreement between the independent reviewers is not reported in your results.5. Your review cannot be both narrative and systematic, you may decide what it is and use the term consistently.6. How did you calculate the missing means before pooling outcomes?7. Was the clinical measurements comparable across studies? Probes? Clinicians? Specialists? Private practice/uni? Calibrations?7. I would like to see in the discussion a comparison of the link of periodontitis with other general conditions and how it compares with SS. What are the effect sizes there? The shown association could very well  be spurious.8. You mention that the etiology of perio in SS is  multifactorial (?) which is not very accurate. This is true for periodontitis in general. Also SS is one variant of sclerosis which is important for the reader to know along with some more information on the condition. 9. I would like to see discussion on the potential direction of the etiological link--in my view it would be more likely that the SS can burden periodontal status for practical but 10. Generally, I would phrase very carefully and tone down the possible link in view of the very sample sizes of the included studies.

Reviewer #2: Interesting and well executed study, although it exhibits the inherent problems when a meta-analysis includes observational studies. Only a minor comment:

- Please perform a thorough english language check as many typo and grammatical errors exist.

6. PLOS authors have the option to publish the peer review history of their article (what does this mean?). If published, this will include your full peer review and any attached files.

Reviewer #1: No

Reviewer #2: No

---

## [Author Response · Author response to Decision Letter 0]

25 Jul 2023

Thank you for allowing us to submit a revised version of our manuscript titled "Periodontal health status in systemic sclerosis patients: systematic review and meta-analysis" (PONE-D-23-12300) to PLOS ONE. We are grateful for the valuable comments provided by the reviewers. We have taken great care to incorporate the suggested changes in order to address the reviewers' recommendations. The revised portions of the manuscript are highlighted in yellow, and detailed explanations of the modifications have been provided in our responses.

---

## [Decision Letter · Decision Letter 1]

22 Aug 2023

Periodontal health status in systemic sclerosis patients: systematic review and meta-analysis

PONE-D-23-12300R1

Dear Dr. Sredojevic,

We’re pleased to inform you that your manuscript has been judged scientifically suitable for publication and will be formally accepted for publication once it meets all outstanding technical requirements.

Kind regards,

Stefano Corbella, DDS, PhD

Academic Editor

PLOS ONE

Additional Editor Comments (optional):

Reviewers' comments:

Reviewer's Responses to Questions

**Comments to the Author**

1. If the authors have adequately addressed your comments raised in a previous round of review and you feel that this manuscript is now acceptable for publication, you may indicate that here to bypass the “Comments to the Author” section, enter your conflict of interest statement in the “Confidential to Editor” section, and submit your "Accept" recommendation.

Reviewer #1: All comments have been addressed

Reviewer #2: All comments have been addressed

2. Is the manuscript technically sound, and do the data support the conclusions?

Reviewer #1: Yes

Reviewer #2: Yes

3. Has the statistical analysis been performed appropriately and rigorously? 

Reviewer #1: Yes

Reviewer #2: Yes

4. Have the authors made all data underlying the findings in their manuscript fully available?

Reviewer #1: Yes

Reviewer #2: Yes

5. Is the manuscript presented in an intelligible fashion and written in standard English?

Reviewer #1: Yes

Reviewer #2: Yes

6. Review Comments to the Author

Reviewer #1: The Authors have revised substantially their initial submission.

Some minor suggestions:

You could make the title less wordy and use "periodontal status in systemic sclerosis patients" and "syst review with MA"

I would also tone done the conclusion throughout the text and especially in the abstract using wording such as "seem to suffer" and avoid mentioning on that part the type of the publication (i.e. systematic review with MA)

I believe this work is methodologically sound and reads well.

Congratulations for the hard work.

Reviewer #2: I would like to thank the authors for addressing my comments. I have no further remarks concerning this paper.

7. PLOS authors have the option to publish the peer review history of their article (what does this mean?). If published, this will include your full peer review and any attached files.

Reviewer #1: No

Reviewer #2: No

---

## [Editor Report · Acceptance letter]

29 Aug 2023

PONE-D-23-12300R1 

Periodontal health status in systemic sclerosis patients: systematic review and meta-analysis 

Dear Dr. Sredojevic:

I'm pleased to inform you that your manuscript has been deemed suitable for publication in PLOS ONE. Congratulations! Your manuscript is now with our production department. 

Kind regards, 

on behalf of

Dr. Stefano Corbella 

Academic Editor

PLOS ONE